# Extension of Scattering Power Decomposition to Dual-Polarization Data for Tropical Forest Monitoring

Ryu Sugimoto [1,*], Ryosuke Nakamura [1], Chiaki Tsutsumi [1] and Yoshio Yamaguchi [2]

1. National Institute of Advanced Industrial Science and Technology (AIST), Tokyo 135-0064, Japan
2. Faculty of Engineering, Niigata University, Niigata 950-2181, Japan
* Correspondence: sugimoto.ryu@aist.go.jp

**Abstract:** A new scattering power decomposition method is developed for accurate tropical forest monitoring that utilizes data in dual-polarization mode instead of quad-polarization (POLSAR) data. This improves the forest classification accuracy and helps to realize rapid deforestation detection because dual-polarization data are more frequently acquired than POLSAR data. The proposed method involves constructing scattering power models for dual-polarization data considering the radar scattering scenario of tropical forests (i.e., ground scattering, volume scattering, and helix scattering). Then, a covariance matrix is created for dual-polarization data and is decomposed to obtain three scattering powers. We evaluated the proposed method by using simulated dual-polarization data for the Amazon, Southeast Asia, and Africa. The proposed method showed an excellent forest classification performance with both user's accuracy and producer's accuracy at >98% for window sizes greater than $7 \times 14$ pixels, regardless of the transmission polarization. It also showed a comparable deforestation detection performance to that obtained by POLSAR data analysis. Moreover, the proposed method showed better classification performance than vegetation indices and was found to be robust regardless of the transmission polarization. When applied to actual dual-polarization data from the Amazon, it provided accurate forest map and deforestation detection. The proposed method will serve tropical forest monitoring very effectively not only for future dual-polarization data but also for accumulated data that have not been fully utilized.

**Keywords:** scattering power decomposition; dual-polarization; tropical forests; forest monitoring; PALSAR-2

## 1. Introduction

Reducing deforestation and forest degradation is an effective option for climate change mitigation because forests can absorb and sequester large amounts of greenhouse gas emissions. Tropical forests and savannas in Latin America have the largest share of mitigation potential, followed by those in Southeast Asia and Africa [1]. Globally, the net forest loss rate declined from 7.8 million ha/year in 1990–2000 to 4.7 million ha/year in 2010–2020. Brazil, the Democratic Republic of the Congo (DRC), and Indonesia suffered the largest average annual net losses of forest area between 2010 and 2020 [2]. In 2012, Brazil reduced the deforestation rate in the Amazon by 84% compared to the historical peak in 2004. However, since 2013, the official deforestation rate has increased, especially in 2019 and 2020 [3]. In DRC, the gross forest cover loss from 2000 to 2010 was estimated to be 37,118 km$^2$, or 2.3% of the total forest area in 2000 [4]. In Sumatra and Kalimantan, primary forest loss increased from 2001 to 2012 and gradually decreased afterward through to 2019 [5].

Early warning of deforestation and forest degradation is crucial to protecting and maintaining tropical forests. Remote sensing by optical sensors has been utilized for monitoring forests in these regions (e.g., PRODES in Brazil and Landsat in DRC and Indonesia). Such optical sensors are very effective for monitoring the long-term trend of forest distributions. However, they are often unsuitable for early detection of deforestation

and forest degradation because of the dense cloud cover over tropical forests during the rainy season.

Synthetic aperture radar (SAR) sensors are suitable for monitoring tropical forests because microwaves can penetrate the cloud cover and observe the ground regardless of the weather. The Japan International Cooperation Agency (JICA) and Japan Aerospace Exploration Agency (JAXA) have been operating the JICA–JAXA Forest Early Warning System in the Tropics (JJ-FAST) since November 2016, which provides forest change information of tropical forests every 1.5 months under all weather conditions [6]. JJ-FAST uses SAR time-series data acquired by Phased Array type L-band Synthetic Aperture Radar-2 (PALSAR-2) in dual-polarization (HH/HV) ScanSAR mode with a spatial resolution of 50 m. Global Forest Watch has been operating Radar for Detecting Deforestation (RADD) alerts since 2019 [7], which provides tropical forest disturbance alerts every 6–12 days based on an analysis of SAR time-series data acquired by Sentinel-1 in dual-polarization (VV/VH) mode with a spatial resolution of 10 m. Both JJ-FAST and RADD use algorithms that analyze the backscatter of each polarization and do not utilize the correlation between polarizations. Polarimetric SAR (POLSAR) data are acquired in quad-polarization Stripmap mode, and L-band POLSAR data, in particular, contain the most information regarding the dielectric properties and structures of scatterers. Deforestation detection based on PALSAR-2 POLSAR data from before and after logging and using the six-component scattering power decomposition (6SD) method demonstrated an almost comparable performance with detection based on an annual deforestation map using time-series data from an optical sensor [8]. However, the swath coverage of PALSAR-2 data in Stripmap mode is insufficient for monitoring tropical forests. PALSAR-3 is the next-generation successor of PALSAR-2 and has a spatial resolution of 10 m with a wide swath of 200 km in dual-polarization Stripmap mode [9]. Thus, PALSAR–3 will provide abundant dual-polarization data for monitoring tropical forests. A few studies of the polarimetric analysis for dual-polarization data were conducted. In [10], the entropy/alpha decomposition method for POLSAR data was extended to dual-polarization data. Although the previous study applied the method to several application examples, it did not evaluate the technique quantitatively. It was indicated in [11] that HH/HV and HV/VV data did not adequately extract scattering mechanisms in the entropy/alpha decomposition method.

To realize accurate tropical forest monitoring using dual-polarization data, the objective of this paper is to extend the scattering power decomposition method for POLSAR data to dual-polarization data. We quantitatively evaluated the proposed method by using simulated dual-polarization data from the Amazon, Southeast Asia, and Africa and comparing the forest classification and deforestation detection performance to those of the 6SD method and vegetation indices. We also evaluated its performance when applied to actual dual-polarization data from the Amazon.

## 2. Data and Methods

### 2.1. Data

#### 2.1.1. PALSAR-2 POLSAR Data for Simulating Dual-Polarization Data

PALSAR-2 was launched on 24 May 2014, and it is an Earth observation L-band SAR sensor onboard the Advanced Land Observing Satellite 2 (ALOS-2). PALSAR-2 has three observation modes [12]: Spotlight (single-polarization), Stripmap (single-, dual-, and quad-polarization), and ScanSAR (single- and dual-polarization). In the basic observation scenario of PALSAR-2, the dual-polarization Stripmap and ScanSAR modes cover almost the entire Earth. PALSAR-2 observes the terrestrial areas of Earth twice a year in dual-polarization Stripmap mode with a spatial resolution of 10 m and swath width of 70 km. The dual-polarization ScanSAR mode has a wide swath width of 350 km but a much lower spatial resolution. The quad-polarization Stripmap mode has a spatial resolution of 6 m and swath width of 50 km, which are almost comparable to the spatial resolution and swath width of dual-polarization data. This means that HH/HV or VV/VH data in POLSAR data can be used to simulate dual-polarization data and validate the proposed method

that extends the scattering power decomposition method to the dual-polarization data. However, few time-series data have been acquired in quad-polarization mode, even during the global basic observation scenario for its mode that ended in June 2017. Rio Branco in Brazil is one of the few sites for which time series POLSAR data are available because corner reflectors were installed here for the PALSAR-2 calibration and validation [13]. Thus, POLSAR data for this region have been acquired at least once a year since 2017. Furthermore, Rio Branco is at the border between Peru, Bolivia, and Brazil, and it is at the western terminal of the area with the highest rate of deforestation in the Amazon, which is known as the "Arc of Deforestation." Therefore, we selected Rio Branco as a study site, as shown in Figure 1. In addition, we selected three other study sites for which POLSAR data of tropical forests have been acquired: the Amazon, Southeast Asia, and Africa. The Amazon study site is around the Ucayali River and it includes a flooded area. The co-polarization (HH or VV) component is greater in flooded forests than in non-flooded forests owing to the dihedral reflection of stems [6,8] and flooded forests can include a backscatter variation in HH polarization exceeding 3 dB compared to non-flooded forests [14]. The Kalimantan study site in Southeast Asia includes artificial forests such as oil palm plantations, which have different scattering behavior than natural vegetation [8]. Oil palm plantations increase the double-bounce scattering power in addition to the volume scattering power for vegetation because L-band microwaves can penetrate the canopies of oil palms to interact with the woody trunks and the underlying surface. The Africa study site covers the Congo Basin, which has many tropical forests. Table 1 presents the PALSAR-2 data used in this study. The data acquired at Rio Branco were used to produce the forest map of each scene and detect deforestation during the three years from 2015 to 2017. Other data were used to produce the forest map of each study site.

**Table 1.** Main characteristics of POLSAR data from PALSAR-2 used in this study.

| Site | Subarea | Off-Nadir Angle (°) | Acquisition Date |
|---|---|---|---|
| Rio Branco | A | 28.4° | 9 January 2015 19 January 2018 |
| | B | 30.9° | 23 January 2015 5 January 2018 |
| Ucayali River | – | 30.9° | 16 April 2016 |
| Kalimantan | A | 33.2° | 9 January 2016 29 October 2016 |
| | B | 28.4° | 8 December 2014 27 April 2015 |
| Congo Basin | A | 28.4° | 8 November 2014 7 May 2016 |
| | B | | 8 November 2014 7 May 2016 |

2.1.2. Reference Data

Visual interpretation of images with a high spatial resolution is often utilized to construct reference data to validate the image processing results of satellite data [4,6–8]. At Rio Branco, reference data were produced by visual interpretation of an area comprising 67.2°–67.5°W and 9.5°–9.8°S using the Planet biannual base map of the NICFI satellite data program [8]. The reference data consisted of polygons representing deforestation, permanent forest, and permanent non-forest sites in Rio Branco between January 2015 and January 2018. For the reference forest map in January 2015, the deforestation and permanent forest polygons could be utilized. In addition, for the reference forest map in January 2018, only permanent forest polygons could be utilized. The reference data were used to validate the results at Rio Branco considering the above relationship. Note that the deforestation polygons of 1 ha or less were also used, which we excluded in a previous

study due to target size [8]. Those polygons were then rasterized with a spatial resolution of 1 arcsec, which had the same spatial resolution as that of the digital elevation model we used. The raster images indicated that the deforested sites in the study area covered ~2000 ha in total, as given in Table 2. The permanent forest and permanent non-forest sites each covered areas of ~30,000 ha.

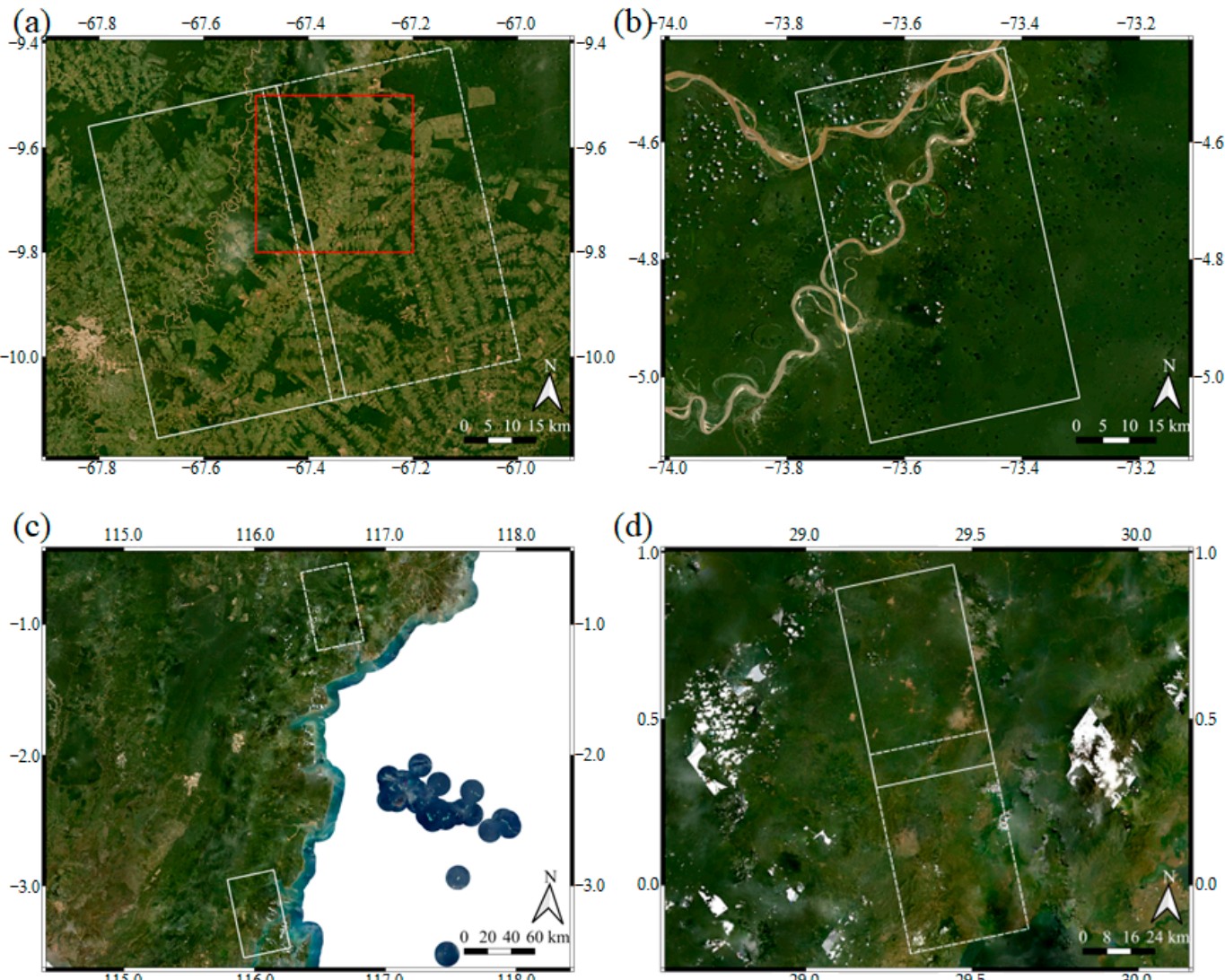

**Figure 1.** Study sites: (**a**) Rio Branco in Brazil, (**b**) Ucayali River in the Amazon, (**c**) Kalimantan in Indonesia, and (**d**) Congo Basin in the Republic of the Congo. The white solid and dotted rectangles indicate subareas A and B, respectively. The red square in (**a**) shows the reference data area described in Section 2.1.2. The mosaic images used as the base map were acquired by Planet/Dove in (**a**) December 2017–May 2018, (**b**) June–November 2016, (**c**) June–November 2016, and (**d**) December 2015–May 2016.

**Table 2.** Number of polygons, pixels, and corresponding area for deforestation, permanent forest, and permanent non-forest sites at Rio Branco.

|  | Polygons | Pixels | Area (ha) |
|---|---|---|---|
| Deforestation | 406 | 21,281 | 1915 |
| Permanent forest | 514 | 341,355 | 30,722 |
| Permanent non-forest | 648 | 394,589 | 35,513 |

However, no reference data by visual interpretation were available for the other study sites. The objective of our study was to extend an analysis method for POLSAR data to dual-polarization data. Therefore, we decided to use the POLSAR results as reference data for the Ucayali River, Kalimantan, and Congo Basin sites and validated the results using dual-polarization data against the consistency with the POLSAR data.

*2.2. Proposed Method*

2.2.1. Scattering Power Decomposition

The scattering power decomposition algorithm is part of the POLSAR analyses, and several types have been proposed. The three-component scattering power decomposition method [15] considers a simple physical scattering model (i.e., surface scattering, double-bounce scattering, and volume scattering), and decomposes POLSAR data under reflection symmetry conditions. This method is suitable for natural distributed targets. Singh et al. [16] proposed a method called general four-component decomposition with a unitary transformation of the coherency matrix (G4U). Singh and Yamaguchi [17] later extended the G4U method to include two additional scattering models (i.e., oriented and compound dipole) into the 6SD method for better physical interpretation. The 6SD method divides the total scattering power into six components: the surface scattering power $P_s$, double-bounce scattering power $P_d$, volume scattering power $P_v$, helix scattering power $P_h$, oriented dipole scattering power $P_{od}$, and compound dipole scattering power $P_{cd}$. It considerably improves image interpretation compared with other existing model-based decompositions [17]. A scattering power decomposition algorithm is generally expressed as follows:

$$\langle [T] \rangle = \langle [R_p(\theta)] k_p k_p^\dagger [R_p(\theta)]^\dagger \rangle = \langle k_p(\theta) k_p^\dagger(\theta) \rangle = \int_0^{2\pi} k_p(\theta) k_p^\dagger(\theta) p(\theta) d\theta \\ = \sum P_{comp} \langle [T] \rangle_{comp} \tag{1}$$

$$k_p = \frac{1}{\sqrt{2}} \begin{bmatrix} S_{HH} + S_{VV} \\ S_{HH} - S_{VV} \\ 2S_{HV} \end{bmatrix} \tag{2}$$

$$R_p(\theta) = \begin{bmatrix} 1 & 0 & 0 \\ 0 & \cos 2\theta & \sin 2\theta \\ 0 & -\sin 2\theta & \cos 2\theta \end{bmatrix} \tag{3}$$

where $\langle [T] \rangle$ is a $3 \times 3$ scattering coherency matrix with an ensemble average and $k_p$ is the Pauli scattering vector. $\theta$ is the polarimetric orientation angle and $p(\theta)$ is a probability density function that represented its distribution. $P_{comp}$ and $\langle [T] \rangle_{comp}$ are components of scattering powers and expansion matrices corresponding to scattering powers such as the surface scattering power and double-bounce scattering power. $S_{HH}$, $S_{VV}$, and $S_{HV}$ are complex elements of the scattering matrix. Refer to [18] for further details on expansion matrices corresponding to scattering powers.

2.2.2. Extension to Dual-Polarization Data

The Pauli scattering vector and coherency matrix cannot be constructed for dual-polarization data because only two complex elements are acquired: $S_{HH}$, $S_{HV}$ or $S_{VV}$, $S_{VH}$. Therefore, we cannot directly apply scattering power decomposition algorithms intended for POLSAR data such as G4U and 6SD to dual-polarization data. Based on the relationship between dual-polarization and POLSAR data, we can summarize the polarimetric information content of the $2 \times 2$ covariance matrices $[C2_H]$, $[C2_V]$ by relating them to the full $3 \times 3$ scattering coherency matrices as follows [10]:

$$k_H = \begin{bmatrix} S_{HH} \\ S_{HV} \end{bmatrix} = \frac{1}{2} \begin{bmatrix} 1 & 1 & 0 \\ 0 & 0 & 1 \end{bmatrix} \begin{bmatrix} S_{HH} + S_{VV} \\ S_{HH} - S_{VV} \\ 2S_{HV} \end{bmatrix} = \frac{1}{\sqrt{2}} \begin{bmatrix} 1 & 1 & 0 \\ 0 & 0 & 1 \end{bmatrix} k_p \tag{4}$$

$$[C2_H] = k_H k_H^\dagger = \begin{bmatrix} |S_{HH}|^2 & S_{HH} S_{HV}^* \\ S_{HH}^* S_{HV} & |S_{HV}|^2 \end{bmatrix} = \frac{1}{2} \begin{bmatrix} 1 & 1 & 0 \\ 0 & 0 & 1 \end{bmatrix} k_p k_p^\dagger \begin{bmatrix} 1 & 0 \\ 1 & 0 \\ 0 & 1 \end{bmatrix} \tag{5}$$

$$k_V = \begin{bmatrix} S_{VV} \\ S_{VH} \end{bmatrix} = \frac{1}{2} \begin{bmatrix} 1 & -1 & 0 \\ 0 & 0 & 1 \end{bmatrix} \begin{bmatrix} S_{HH} + S_{VV} \\ S_{HH} - S_{VV} \\ 2S_{HV} \end{bmatrix} = \frac{1}{\sqrt{2}} \begin{bmatrix} 1 & -1 & 0 \\ 0 & 0 & 1 \end{bmatrix} k_p \tag{6}$$

$$\because S_{VH} = S_{HV}$$

$$[C2_V] = k_V k_V^\dagger = \begin{bmatrix} |S_{VV}|^2 & S_{VV}S_{VH}^* \\ S_{VV}^* S_{VH} & |S_{VH}|^2 \end{bmatrix} = \frac{1}{2}\begin{bmatrix} 1 & -1 & 0 \\ 0 & 0 & 1 \end{bmatrix} k_p k_p^\dagger \begin{bmatrix} 1 & 0 \\ -1 & 0 \\ 0 & 1 \end{bmatrix}. \tag{7}$$

These relationships can then be used to extend a scattering power decomposition algorithm to dual-polarization data. Therefore, we construct three scattering power models for dual-polarization data considering the radar scattering scenario of tropical forests: volume scattering power, helix scattering power, and ground scattering power.

**Volume scattering power**

A cloud of randomly oriented dipole is employed as the volume scattering model because its power is mainly attributed to natural vegetation with a random branch distribution. The Pauli scattering vector $k_p(\theta)$ of the dipole model [18] and the covariance matrices can be expressed as follows:

$$k_p(\theta) = \begin{bmatrix} 1 & \cos 2\theta & -\sin 2\theta \end{bmatrix}^T$$
$$\langle [C2_H] \rangle_v = \int_0^{2\pi} \frac{1}{2}\begin{bmatrix} 1 + 2\cos 2\theta + \cos^2 2\theta & -\sin 2\theta(1 + \cos 2\theta) \\ -\sin 2\theta(1 + \cos 2\theta) & \sin^2 2\theta \end{bmatrix} p(\theta)d\theta$$
$$\langle [C2_V] \rangle_v = \int_0^{2\pi} \frac{1}{2}\begin{bmatrix} 1 - 2\cos 2\theta + \cos^2 2\theta & -\sin 2\theta(1 - \cos 2\theta) \\ -\sin 2\theta(1 - \cos 2\theta) & \sin^2 2\theta \end{bmatrix} p(\theta)d\theta \tag{8}$$
$$\text{(for a horizontal dipole)}$$

$$k_p(\theta) = \begin{bmatrix} 1 & -\cos 2\theta & \sin 2\theta \end{bmatrix}^T$$
$$\langle [C2_H] \rangle_v = \int_0^{2\pi} \frac{1}{2}\begin{bmatrix} 1 - 2\cos 2\theta + \cos^2 2\theta & \sin 2\theta(1 - \cos 2\theta) \\ \sin 2\theta(1 - \cos 2\theta) & \sin^2 2\theta \end{bmatrix} p(\theta)d\theta$$
$$\langle [C2_V] \rangle_v = \int_0^{2\pi} \frac{1}{2}\begin{bmatrix} 1 + 2\cos 2\theta + \cos^2 2\theta & \sin 2\theta(1 + \cos 2\theta) \\ \sin 2\theta(1 + \cos 2\theta) & \sin^2 2\theta \end{bmatrix} p(\theta)d\theta. \tag{9}$$
$$\text{(for a vertical dipole)}$$

Uniform, vertical, and horizontal distributions are generally considered for the probability density function of a dipole model. Both co-polarization data ($S_{HH}$ and $S_{VV}$) are needed to distinguish these distributions. If the probability density function of the dipole model has the uniform distribution $p(\theta) = 1/2\pi$, then the covariance matrix of the volume scattering power finally becomes:

$$\langle [C2_H] \rangle_v = \langle [C2_V] \rangle_v = \frac{1}{4}\begin{bmatrix} 3 & 0 \\ 0 & 1 \end{bmatrix}. \tag{10}$$
$$\text{(for both horizontal and vertical dipole)}$$

When the probability density function is not a uniform distribution, the scatters are primarily distributed vertically or horizontally, and the covariance matrix differs from that in Equation (10). By using an appropriately large window size for the ensemble average, we can assume that the probability density function has a uniform distribution.

**Helix scattering power**

The left or right helix model is the only model that can account for the off-diagonal term in the covariance matrix (i.e., $S_{HH}S_{HV}^* \neq 0$, $S_{VV}S_{VH}^* \neq 0$) [18], and the helix target generates circular polarization for all linear polarization incidences. Then, the Pauli scattering vector $k_p(\theta)$ of the helix model and the covariance matrices are expressed as follows:

$$k_p(\theta) = e^{j2\theta}\begin{bmatrix} 0 & 1 & j \end{bmatrix}^T$$
$$\langle [C2_H] \rangle_h = \int_0^{2\pi} \frac{1}{2}\begin{bmatrix} 1 & -j \\ j & 1 \end{bmatrix} p(\theta)d\theta$$
$$\langle [C2_V] \rangle_h = \int_0^{2\pi} \frac{1}{2}\begin{bmatrix} 1 & j \\ -j & 1 \end{bmatrix} p(\theta)d\theta \tag{11}$$
$$\text{(for a left helix)}$$

$$k_p(\theta) = e^{-j2\theta}\begin{bmatrix} 0 & 1 & -j \end{bmatrix}^T$$
$$\langle [C2_H] \rangle_h = \int_0^{2\pi} \frac{1}{2}\begin{bmatrix} 1 & j \\ -j & 1 \end{bmatrix} p(\theta)d\theta$$
$$\langle [C2_V] \rangle_h = \int_0^{2\pi} \frac{1}{2}\begin{bmatrix} 1 & -j \\ j & 1 \end{bmatrix} p(\theta)d\theta. \tag{12}$$
$$\text{(for a right helix)}$$

The covariance matrix of the helix scattering power is independent of the probability density function and is finally summarized as follows:

$$\langle[C2_H]\rangle_h = \frac{1}{2}\begin{bmatrix} 1 & -j \\ j & 1 \end{bmatrix}, \ \langle[C2_V]\rangle_h = \frac{1}{2}\begin{bmatrix} 1 & j \\ -j & 1 \end{bmatrix}$$
$$\text{(for a left helix)}$$
$$\langle[C2_H]\rangle_h = \frac{1}{2}\begin{bmatrix} 1 & j \\ -j & 1 \end{bmatrix}, \ \langle[C2_V]\rangle_h = \frac{1}{2}\begin{bmatrix} 1 & -j \\ j & 1 \end{bmatrix}.$$
$$\text{(for a right helix)}$$
(13)

**Ground scattering power**

The scattering components on the ground are mainly surface scattering and double-bounce scattering, except for volume scattering. Those components approximately correspond to $\langle|S_{HH} + S_{VV}|^2\rangle$ and $\langle|S_{HH} - S_{VV}|^2\rangle$, respectively, and are distinguished by the phase difference between $S_{HH}$ and $S_{VV}$ (i.e., it is in phase for surface scattering and vice versa for double-bounce scattering). However, $S_{HH}$ and $S_{VV}$ cannot be acquired simultaneously in dual-polarization mode. Because the left-upper diagonal terms in the covariance matrix of dual-polarization data are $S_{HH}$ and $S_{VV}$, we can express the ground scattering power as follows:

$$\langle[C2_H]\rangle_g = \langle[C2_V]\rangle_g = \begin{bmatrix} 1 & 0 \\ 0 & 0 \end{bmatrix}.$$
(14)

**Scattering power decomposition for dual-polarization data**

A covariance matrix acquired in dual-polarization mode can be expressed by using three unknown scattering powers ($P_g^H$, $P_v^H$, and $P_h^H$ for horizontal transmission polarization or $P_g^V$, $P_v^V$, and $P_h^V$ for vertical transmission polarization) and Equations (10), (13), and (14) as follows:

$$[C2_H] = P_g^H\langle[C2_H]\rangle_g + P_v^H\langle[C2_H]\rangle_v + P_h^H\langle[C2_H]\rangle_h$$
$$= P_g^H\begin{bmatrix} 1 & 0 \\ 0 & 0 \end{bmatrix} + \frac{P_v^H}{4}\begin{bmatrix} 3 & 0 \\ 0 & 1 \end{bmatrix} + \frac{P_h^H}{2}\begin{bmatrix} 1 & j \\ -j & 1 \end{bmatrix}$$
(15)

$$[C2_V] = P_g^V\langle[C2_V]\rangle_g + P_v^V\langle[C2_V]\rangle_v + P_h^V\langle[C2_V]\rangle_h$$
$$= P_g^V\begin{bmatrix} 1 & 0 \\ 0 & 0 \end{bmatrix} + \frac{P_v^V}{4}\begin{bmatrix} 3 & 0 \\ 0 & 1 \end{bmatrix} + \frac{P_h^V}{2}\begin{bmatrix} 1 & -j \\ j & 1 \end{bmatrix}$$
(16)

$$\langle|S_{HH}|^2\rangle = P_g^H + \frac{3}{4}P_v^H + \frac{1}{2}P_h^H, \ \ \langle|S_{VV}|^2\rangle = P_g^V + \frac{3}{4}P_v^V + \frac{1}{2}P_h^V$$
(17)

$$|Im\langle S_{HH}S_{HV}^*\rangle| = \frac{1}{2}P_h^H, \ \ |Im\langle S_{VV}S_{VH}^*\rangle| = \frac{1}{2}P_h^V$$
(18)

$$\langle|S_{HV}|^2\rangle = \frac{1}{4}P_v^H + \frac{1}{2}P_h^H, \ \ \langle|S_{VH}|^2\rangle = \frac{1}{4}P_v^V + \frac{1}{2}P_h^V.$$
(19)

The three unknowns ($P_g^H$, $P_v^H$, and $P_h^H$ for horizontal transmission polarization or $P_g^V$, $P_v^V$, and $P_h^V$ for vertical transmission polarization) can be determined as follows. First, the helix scattering power $P_h$ is determined directly from Equation (18):

$$P_h^H = 2|Im\langle S_{HH}S_{HV}^*\rangle|, P_h^V = 2|Im\langle S_{VV}S_{VH}^*\rangle|.$$
(20)

Then, Equation (19) gives the volume scattering power $P_v$:

$$P_v^H = 4\langle|S_{HV}|^2\rangle - 2P_h^H, P_v^V = 4\langle|S_{VH}|^2\rangle - 2P_h^V.$$
(21)

Finally, the ground scattering power $P_g$ is determined from Equation (17):

$$P_g^H = \langle|S_{HH}|^2\rangle - \frac{3}{4}P_v^H - \frac{1}{2}P_h^H = TP^H - P_v^H - P_h^H,$$
$$P_g^V = \langle|S_{VV}|^2\rangle - \frac{3}{4}P_v^V - \frac{1}{2}P_h^V = TP^V - P_v^V - P_h^V$$
(22)

where $TP^H$ and $TP^V$ are the total powers $\langle|S_{HH}|^2\rangle + \langle|S_{HV}|^2\rangle$ and $\langle|S_{VV}|^2\rangle + \langle|S_{VH}|^2\rangle$, respectively.

Thus, the proposed method is a three-component decomposition scheme for using dual-polarization data to monitor forest scattering scenarios. The scattering powers $P_g$, $P_v$, and $P_h$ can be used to monitor and classify radar scenes of tropical forests in the rainy season. Note that the

proposed method has no assumption of applied SAR frequency and is applicable to not only L-band SAR data but also to C-band SAR data. In this study, the proposed method was applied to L-band SAR data with good penetration for vegetation.

*2.3. Validation*

2.3.1. Comparison to 6SD Method at Rio Branco

To evaluate the forest classification accuracy of the proposed method, forest maps were produced using POLSAR data at Rio Branco. First, we used the proposed method to calculate the scattering powers from the simulated dual-polarization data (i.e., HH/HV and VV/VH data in POLSAR data). In scattering power decomposition, the window size of the ensemble average is crucial to distinguish forests [8]. Therefore, we considered various window sizes of $7 \times 14$, $10 \times 20$, and $14 \times 28$ pixels in the range and azimuth directions. These corresponded to ground areas of approximately 0.3, 0.6, and 1 ha, respectively. Next, the scattering powers were geocoded by using Shuttle Radar Topography Mission 1 arc-second Global (SRTM1) for comparison with the reference data projected on the map. The geocoded scattering powers were then averaged over $3 \times 3$ pixels to reduce noise, and those that satisfied the condition $P_v \geq P_g \cap P_v \geq \alpha$ were classified as forests. The classified labels were compared to the reference data (see Section 2.1.2). The threshold $\alpha$ was increased from 0.05 to 0.45 in increments of 0.01 to determine the optimal value for forest classification among the 41 results. The same procedure was performed by using the 6SD method with the POLSAR data for comparison.

Moreover, to evaluate the deforestation detection accuracy with the proposed method, we applied it to POLSAR data for Rio Branco for the 3-year period of 2015–2017. POLSAR data acquired in January 2015 and January 2018 were used to represent before and after logging, respectively. The geocoded scattering powers were calculated in the same manner as for the forest classification accuracy. Then, pixels that satisfied the following condition were classified as deforestation:

$$P_{v\,before} \geq P_{g\,before} \cap P_{v\,before} \geq \alpha \cap P_{v\,after} < \alpha \cap \left( P_{v\,after} - P_{v\,before} \right) < \beta \tag{23}$$

where "before" and "after" indicate before and after logging, respectively. Equation (23) corresponds to Equation (6) in a previous paper [8]. The optimal values of the parameters $\alpha$ and $\beta$ for deforestation detection were determined among the 656 results by increasing $\alpha$ from 0.05 to 0.45 in increments of 0.01 and increasing $\beta$ from $-0.15$ to 0.0 in increments of 0.01. The same procedure was used to evaluate the deforestation detection performance of the 6SD method with the POLSAR data for comparison with the proposed method.

2.3.2. Comparison to 6SD Method at Other Study Sites

A previous study [8] has shown that the 6SD method can robustly distinguish between flooded and non-flooded forests as well as oil palm plantations and natural vegetation. We evaluated whether the proposed method has similar robustness by using POLSAR data acquired at the Ucayali River and Kalimantan. POLSAR data acquired at the Congo Basin were also used to evaluate the robustness of the regional characteristics. Forest maps with slant-range coordinates were produced by using the proposed and 6SD method. The results of the 6SD method were used as reference data because no reference data from visual interpretation were available for these study sites. Thus, we evaluated the forest map generated by the proposed method for its consistency with that generated by the 6SD method. Scattering powers were calculated by using a window size of $10 \times 20$ pixels in the range and azimuth directions. In addition, the scattering powers were decimated to be a spatial resolution equivalent to that of SRTM1 in the slant-range coordinates. The scattering powers were then averaged over areas of $3 \times 3$ pixels to reduce noise, and pixels that satisfied the condition $P_v \geq P_g \cap P_v \geq \alpha$ were classified as forests. For the forest map generated by the 6SD method, the threshold $\alpha$ was determined to be 0.20 based on rough visual interpretation so that the forest and non-forest areas in the 6SD RGB image roughly corresponded to the mosaic image acquired by Planet/Dove rather than pixel-by-pixel. The mosaic images used for the rough visual interpretation were acquired around the same dates as the POLSAR data, except for the mosaic image of the Ucayali River because of dense cloud cover. For the forest map generated by the proposed method, the threshold $\alpha$ was increased from 0.05 to 0.25 in increments of 0.01 to determine the optimal value that best matched the results of the 6SD method, among the 21 results.

2.3.3. Comparison to Vegetation Indices

Table 3 presents various studies that have proposed different methods of using dual-polarization data for forest classification. The radar forest degradation index (RFDI) [19] uses the normalized

differential index for deforestation detection. RFDI generates values between –1 and +1, and it can be seen from Equation (10) that the theoretical value is 0.5 for a forest. The radar vegetation index (RVI) [20] can be applied to dual-polarization data under the assumption $|S_{HH}|^2 \approx |S_{VV}|^2$ [21]. We checked the forest pixels in the reference data to confirm that this assumption was appropriate. RVI generally ranges between 0 and 1, and it can be seen from Equation (10) that the theoretical value is 1.0 for a forest. Forest maps were generated using the RFDI and RVI at the Rio Branco. Based on the value range and the theoretical value of both indices, we classified pixels as forests under the condition $\alpha 1 \leq RFDI \cap RFDI \leq \alpha 2$, and $\alpha 1 \leq RVI$, respectively. To obtain the best forest classification performances with the RFDI and RVI, the thresholds $\alpha 1$ and $\alpha 2$ for the RFDI were evaluated from 0.2 to 0.8 in increments of 0.01, and the threshold $\alpha 1$ for the RVI was evaluated from 0.5 to 1.0 in increments of 0.01. Note that pixels with $|S_{HH}|^2$ and $|S_{VV}|^2$ values lower than 0.03 were considered water areas and classified as non-forest.

**Table 3.** Vegetation indices proposed in previous studies.

| Symbol | Description | Equation | Theoretical Value for Forest |
|---|---|---|---|
| RFDI | Radar forest degradation index | $\dfrac{\|S_{HH}\|^2 - \|S_{HV}\|^2}{\|S_{HH}\|^2 + \|S_{HV}\|^2}$ | 0.5 |
| RVI | Radar vegetation index | $\dfrac{8\|S_{HV}\|^2}{\|S_{HH}\|^2 + \|S_{VV}\|^2 + 2\|S_{HV}\|^2}$ $= \dfrac{4\|S_{HV}\|^2}{\|S_{HH}\|^2 + \|S_{HV}\|^2}$ $\because \|S_{HH}\|^2 \approx \|S_{VV}\|^2 \ at \ forests$ | 1.0 |

### 2.3.4. Application to Actual Dual-Polarization Data

Finally, the proposed method was applied to actual dual-polarization data. In February 2022, JICA conducted a field survey of deforestation at Altamira in Brazil. Therefore, we selected Altamira as the study site and collected dual-polarization data after February 2022 to represent the site after logging. To represent the site before logging, we collected data near the beginning of the year for comparison with the JAXA Global PALSAR-2 Forest/Non-Forest map (JAXA FNF), which is produced annually [22].

## 3. Results

### 3.1. Comparison to 6SD Method at Rio Branco

### 3.1.1. Forest Classification Accuracy

Figure 2 shows the RGB image and the forest map generated by the proposed method at Rio Branco using HH/HV data, a window size of $10 \times 20$ pixels, and a threshold $\alpha$ value of 0.16 as given in Table 4. Pixels dominated by the volume scattering power caused by natural forests are shown in green in the RGB image. The green pixels in the RGB image and forest map are roughly consistent with the visual interpretation of the Planet/Dove image. Table 4 presents the forest classification performances of the proposed method and 6SD method. For comparison of the proposed method with the 6SD method, each best performance by a simple threshold value was demonstrated using the whole reference data. The proposed method showed an excellent classification performance with both the user's accuracy (UA) and producer's accuracy (PA) above 98% regardless of the ensemble average window size and transmission polarization. The high Kappa value of >0.98 indicates that the coincidence was not by chance. The proposed method performed comparably to the 6SD method. Note that only one co-polarization component is included in dual-polarization data, and the total power is lower than that for the POLSAR data, which explains the lower threshold for the proposed method than for the 6SD method.

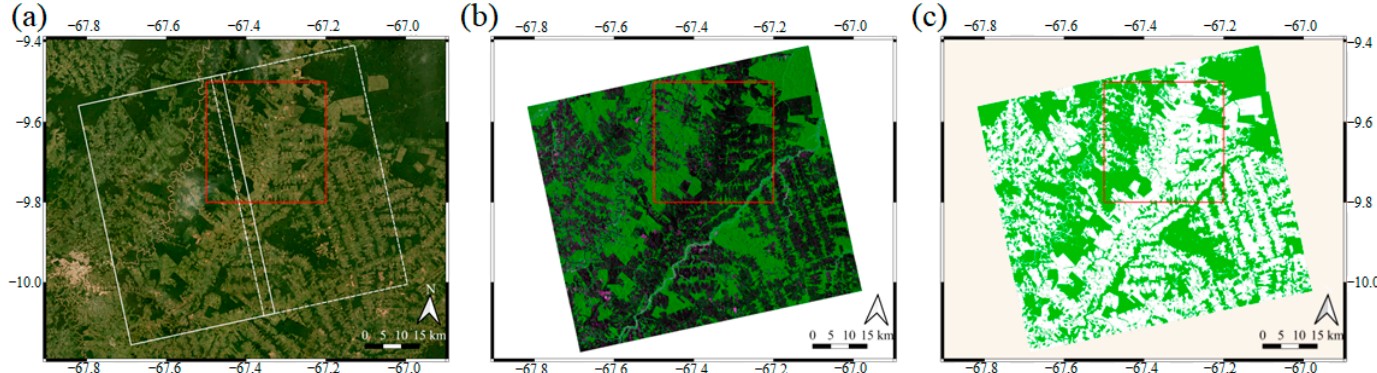

**Figure 2.** Forest map at Rio Branco generated by using the proposed method with HH/HV data acquired on 5 January and 19, 2018, and a window size of 10 × 20 pixels. The red rectangle shows the reference data area. (**a**) Mosaic image acquired by Planet/Dove from December 2017 to May 2018. The white rectangle shows the observation area of PALSAR-2. (**b**) RGB image generated by the proposed method: ground scattering is in red and blue and volume scattering is in green. (**c**) Forest map generated by the proposed method.

**Table 4.** Forest classification performance at Rio Branco.

| Method and Data | Window Size for Ensemble Average | Threshold $\alpha$ | UA (%) | PA (%) | Kappa |
|---|---|---|---|---|---|
| Proposed method with HH/HV | 7 × 14 pixels | 0.15 | 98.6 | 99.3 | 0.981 |
| | 10 × 20 pixels | 0.16 | 98.6 | 99.5 | 0.983 |
| | 14 × 28 pixels | 0.17 | 98.8 | 99.5 | 0.984 |
| Proposed method with VV/VH | 7 × 14 pixels | 0.15 | 98.6 | 99.3 | 0.980 |
| | 10 × 20 pixels | 0.16 | 98.6 | 99.4 | 0.982 |
| | 14 × 28 pixels | 0.17 | 98.8 | 99.4 | 0.983 |
| 6SD method with POLSAR | 7 × 14 pixels | 0.20 | 98.5 | 99.2 | 0.979 |
| | 10 × 20 pixels | 0.24 | 98.6 | 99.6 | 0.982 |
| | 14 × 28 pixels | 0.28 | 98.8 | 99.5 | 0.984 |

UA: User's accuracy (precision) and PA: Producer's accuracy (recall).

### 3.1.2. Deforestation Detection Accuracy

Table 5 presents the deforestation detection performances of the proposed method and 6SD method. The UA, PA, and Kappa values of the two methods were comparable regardless of the ensemble average window size and transmission polarization. The deforestation maps are not shown here, but they indicated that the trends for the correct detection, omission error, and commission error were the same for both methods. We previously showed that the 6SD method had almost the same deforestation detection accuracy as the deforestation map produced annually from time-series optical satellite imagery [8]. Therefore, the proposed method should also contribute to rapid deforestation detection with the same accuracy as the annual deforestation map because dual-polarization data are observed more frequently. The omission error increased for both the proposed method and 6SD method because sparse trees remained after logging (i.e., forest degradation), and the volume scattering power did not decrease. This problem should be addressed in future work.

**Table 5.** Deforestation detection performance at Rio Branco.

| Method and Data | Window Size for Ensemble Average | Threshold | | UA (%) | PA (%) | Kappa |
|---|---|---|---|---|---|---|
| | | $\alpha$ | $\beta$ | | | |
| Proposed method with HH/HV | 7 × 14 pixels | 0.16 | −0.04 | 88.7 | 68.9 | 0.770 |
| | 10 × 20 pixels | 0.17 | −0.04 | 92.1 | 69.9 | 0.789 |
| | 14 × 28 pixels | 0.18 | −0.04 | 93.3 | 71.0 | 0.802 |

**Table 5.** *Cont.*

| Method and Data | Window Size for Ensemble Average | Threshold | | UA (%) | PA (%) | Kappa |
|---|---|---|---|---|---|---|
| | | $\alpha$ | $\beta$ | | | |
| Proposed method with VV/VH | 7 × 14 pixels | 0.16 | −0.04 | 90.2 | 67.3 | 0.765 |
| | 10 × 20 pixels | 0.17 | −0.04 | 93.2 | 68.9 | 0.787 |
| | 14 × 28 pixels | 0.18 | −0.04 | 94.8 | 69.7 | 0.799 |
| 6SD method with POLSAR | 7 × 14 pixels | 0.21 | −0.06 | 83.2 | 67.9 | 0.741 |
| | 10 × 20 pixels | 0.26 | −0.07 | 90.5 | 70.2 | 0.786 |
| | 14 × 28 pixels | 0.30 | −0.07 | 92.3 | 71.8 | 0.803 |

UA: User's accuracy (precision) and PA: Producer's accuracy (recall).

### 3.2. Comparison to 6SD Method at Other Study Sites

Figures 3–5 show images comparing the performances of the proposed method and 6SD method at the Ucayali River, Kalimantan, and the Congo Basin. In the RGB images of the two methods, the green pixels, which indicated predominant volume scattering power, were roughly consistent with the visual interpretation of the Planet/Dove images regardless of the study sites. In particular, Figure 3 shows that the distribution of forest and non-forest areas around the river were almost identical in both RGB images. In Figure 4, the oil palm plantation was not color-coded as green in both RGB images. The blue and red pixels in the 6SD RGB images corresponded to the magenta pixels in the RGB images of the proposed method because the surface and double-bound scattering powers in the 6SD method were treated as the ground scattering power by the proposed method. Moreover, both forest maps mostly matched the pixels classified as forest in the Planet/Dove images, indicating good accuracy. Table 6 presents the consistency between the proposed method and 6SD method at the Ucayali River, Kalimantan, and the Congo Basin. The PA was over 90% regardless of the transmission polarization and study sites, which indicates that the proposed method correctly classified pixels that the 6SD method classified as forest. In general, the UA and Kappa were over 80% and 0.80, respectively, regardless of the transmission polarization and study sites. One outlier was when the proposed method used the VV/VH data at the Ucayali River. In this case, the proposed method incorrectly classified pixels as forest that the 6SD method classified as non-forest, which reduced the Kappa value to below 0.70. This commission error occurred in flooded forest pixels that were red in the 6SD RGB image. These results suggest that the proposed method and 6SD method had comparable forest classification performances across the study sites except for flooded forests using VV/VH data.

**Table 6.** Consistency between the proposed method and 6SD method using a window size of 10 × 20 pixels at Ucayali River, Kalimantan, and the Congo Basin.

| Site | Subarea | Acquisition Date | Threshold $\alpha$ | UA (%) | PA (%) | Kappa |
|---|---|---|---|---|---|---|
| Proposed method with HH/HV | | | | | | |
| Ucayali River | – | 16 April 2016 | 0.14 | 98.3 | 97.1 | 0.867 |
| Kalimantan | A | 9 January 2016 | 0.14 | 92.1 | 89.9 | 0.875 |
| | | 29 October 2016 | 0.13 | 87.4 | 95.4 | 0.859 |
| | B | 8 December 2014 | 0.13 | 95.7 | 98.9 | 0.844 |
| | | 27 April 2015 | 0.13 | 95.6 | 98.8 | 0.848 |
| Congo Basin | A | 8 November 2014 | 0.13 | 94.9 | 98.9 | 0.861 |
| | | 7 May 2016 | 0.13 | 94.6 | 98.4 | 0.865 |
| | B | 8 November 2014 | 0.14 | 85.2 | 91.1 | 0.838 |
| | | 7 May 2016 | 0.13 | 81.1 | 96.4 | 0.843 |

**Table 6.** *Cont.*

| Site | Subarea | Acquisition Date | Threshold $\alpha$ | UA (%) | PA (%) | Kappa |
|---|---|---|---|---|---|---|
| Proposed method with VV/VH | | | | | | |
| Ucayali River | – | 16 April 2016 | 0.16 | 92.2 | 99.5 | 0.682 |
| Kalimantan | A | 9 January 2016 | 0.13 | 86.8 | 95.1 | 0.869 |
| | | 29 October 2016 | 0.13 | 85.4 | 95.6 | 0.842 |
| | B | 8 December 2014 | 0.13 | 94.6 | 98.8 | 0.806 |
| | | 27 April 2015 | 0.13 | 94.1 | 98.9 | 0.801 |
| Congo Basin | A | 8 November 2014 | 0.13 | 94.0 | 99.1 | 0.839 |
| | | 7 May 2016 | 0.13 | 93.8 | 98.3 | 0.845 |
| | B | 8 November 2014 | 0.14 | 81.1 | 90.3 | 0.801 |
| | | 7 May 2016 | 0.13 | 78.1 | 95.3 | 0.814 |

UA: User's accuracy (precision) and PA: Producer's accuracy (recall).

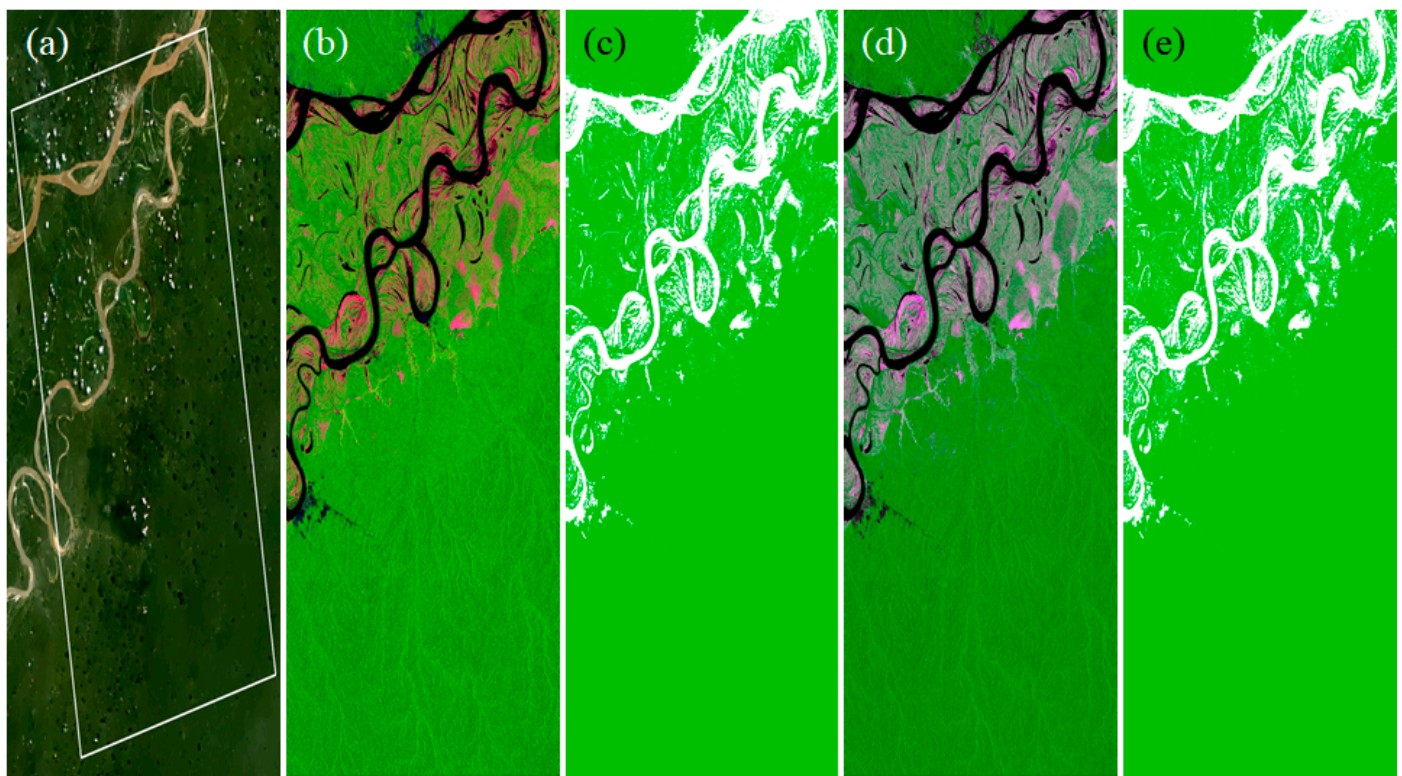

**Figure 3.** Forest maps of Ucayali River in the Amazon using the proposed method and 6SD method with POLSAR data acquired on 16 April 2016 and a window size of $10 \times 20$ pixels. (**a**) Mosaic image acquired by Planet/Dove from June to November 2016. The white rectangle shows the observation area of PALSAR-2. (**b**) 6SD RGB image: double-bounce scattering is in red, volume scattering is in green, and surface scattering is in blue. (**c**) Forest map using the 6SD method. (**d**) RGB image of the proposed method: ground scattering is in red and blue, and volume scattering is in green. (**e**) Forest map using the proposed method.

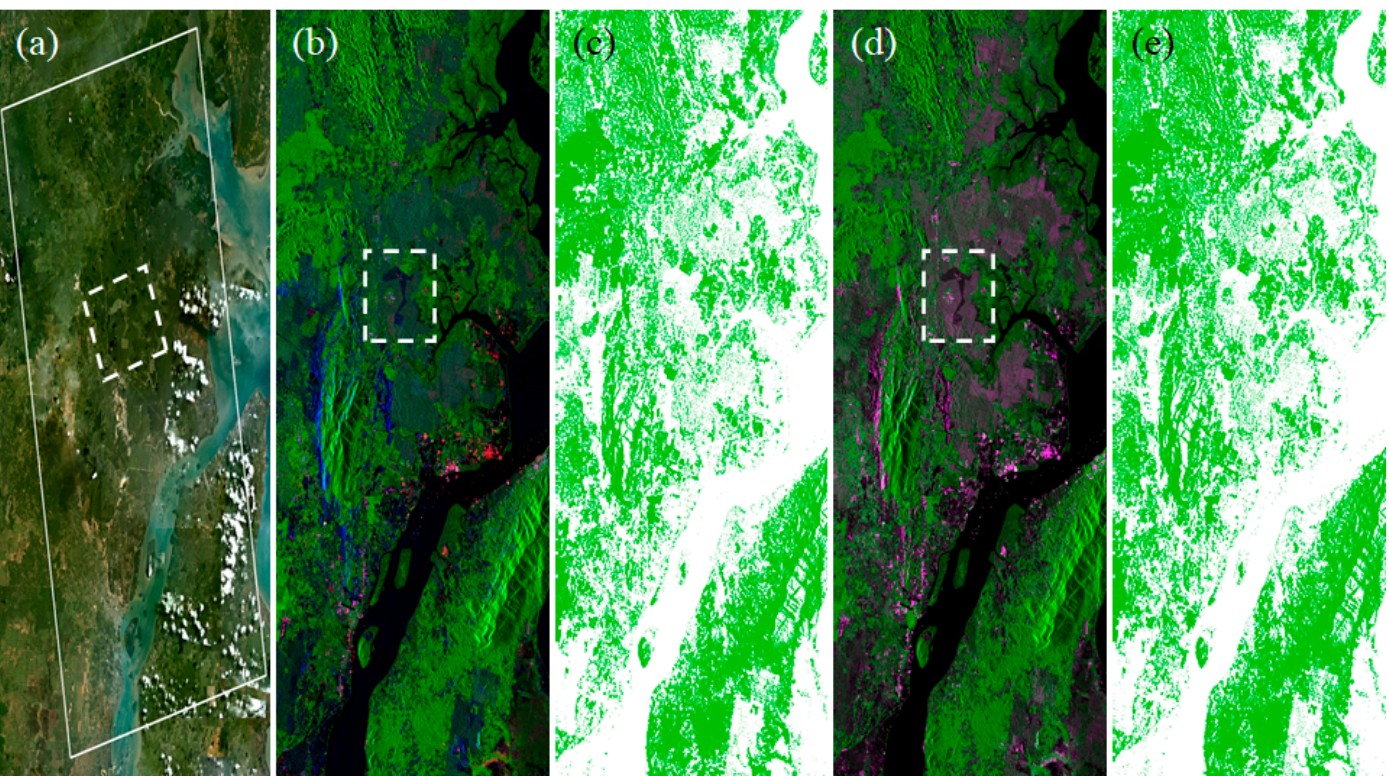

**Figure 4.** Forest maps of Kalimantan in Indonesia using the proposed method and 6SD method with the POLSAR data acquired on 29 October 2016 and a window size of $10 \times 20$ pixels. (**a**) Mosaic image acquired by Planet/Dove from June to November 2016. The white rectangle shows the observation area of PALSAR-2. (**b–e**) Same as in Figure 3. The white dotted rectangle shows the oil palm plantation.

### 3.3. Comparison to Vegetation Indices

Table 7 presents the forest classification performance at Rio Branco using the RFDI and RVI. The reference data are the same ones used in Section 3.1.1 and are summarized in Table 2. The forest classification performances were similar for both indices regardless of the ensemble average window size and transmission polarization. The commission error increased in both indices using VV/VH data. The reason is that the VV power was lower than the HH power in the pixels where the commission error occurred, and both indices resulted in the same value as the forests even though the VH power decreased. From Tables 4 and 7, the proposed method showed higher performance than the RFDI and RVI and had robustness that did not degrade with VV/VH data. Figure 6 shows the forest and non-forest distributions of the volume scattering power derived by the proposed method, RFDI, and RVI using HH/HV data and a window size of $10 \times 20$ pixels. The reference data were used as the forest and non-forest pixels. Among these distributions, the volume scattering power showed a sharp peak at 0.23 and a smaller overlap between forest and non-forest distributions. These distributions support that the proposed method has a superior forest classification performance compared to the vegetation indices.

### 3.4. Application to Actual Dual-Polarization Data

Figure 7 shows the forest map generated at Altamira using the proposed method and JAXA FNF. Note that the forest map generated by the proposed method was based on HH/HV data acquired on April 4, 2019, a window size of $10 \times 20$ pixels, and a threshold $\alpha$ value of 0.16 as given in Table 4. The JAXA FNF was produced in 2018. The forest map generated by the proposed method had less salt-and-pepper noise, and its forest pixels showed good agreement with the forest pixels (>90% crown cover) of JAXA FNF. The non-forest pixels in the forest map generated by the proposed method (red arrows) were confirmed by Planet/Dove mosaic images to be deforested in 2019. The forest pixels with 10–90% crown cover in JAXA FNF were often at the boundary between forests (>90% crown cover) and non-forests, and some were classified as non-forests in the forest map generated by the proposed method. In this case, the percentage of forest in the ensemble average window

decreased, which decreased the volume scattering power. Future work will involve improving the proposed method to clearly distinguish forests with a low canopy cover from non-forests.

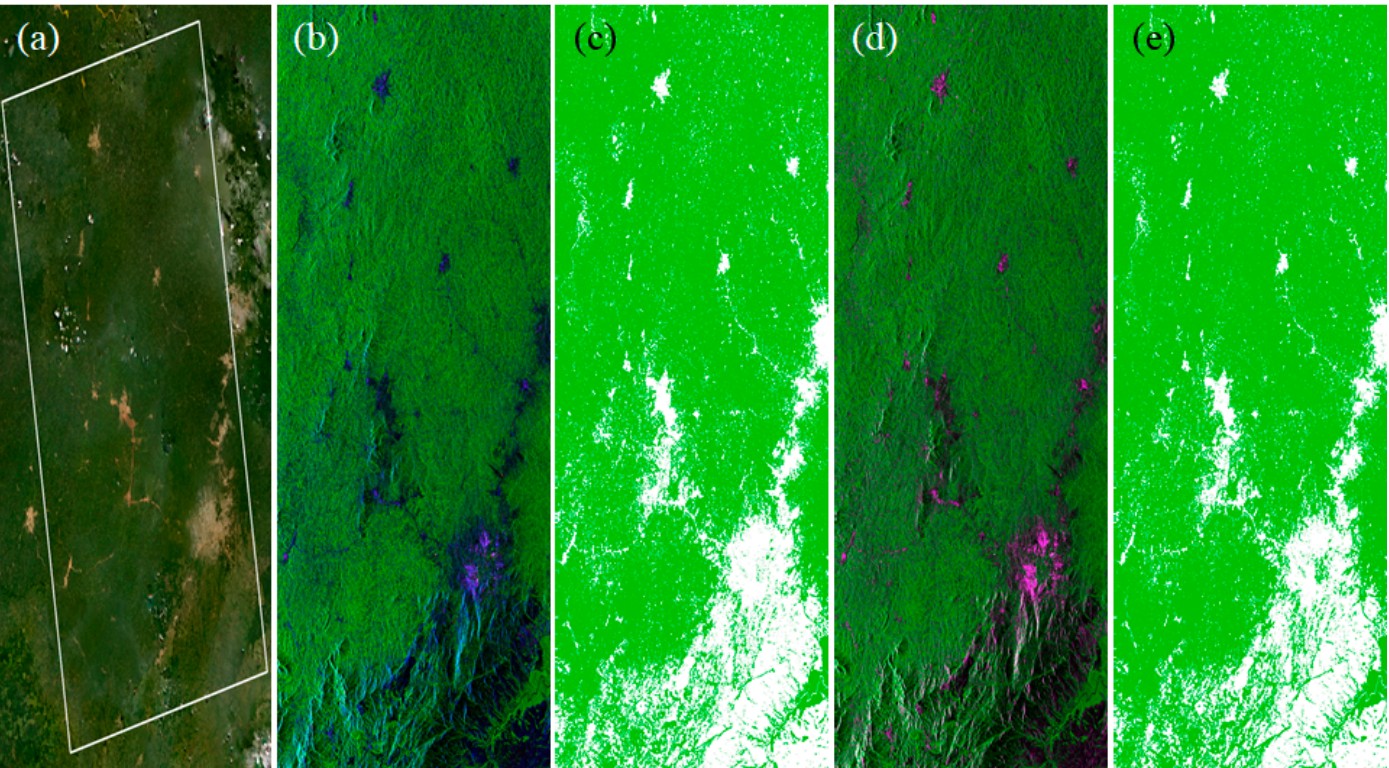

**Figure 5.** Forest maps of the Congo Basin in the Republic of the Congo using the proposed method and 6SD method with the POLSAR data acquired on 7 May 2016 and a window size of 10 × 20 pixels. (**a**) Mosaic image acquired by Planet/Dove from December 2015 to May 2016. The white rectangle shows the observation area of PALSAR-2. (**b**–**e**) Same as in Figure 3.

**Table 7.** Forest classification performance with vegetation indices at Rio Branco.

| Method and Data | Window Size for Ensemble Average | Threshold | | UA (%) | PA (%) | Kappa |
|---|---|---|---|---|---|---|
| | | α1 | α2 | | | |
| RFDI with HH/HV | 7 × 14 pixels | 0.34 | 0.61 | 95.6 | 97.2 | 0.933 |
| | 10 × 20 pixels | 0.34 | 0.61 | 96.0 | 97.7 | 0.940 |
| | 14 × 28 pixels | 0.38 | 0.60 | 97.1 | 97.0 | 0.946 |
| RFDI with VV/VH | 7 × 14 pixels | 0.38 | 0.58 | 86.0 | 96.8 | 0.824 |
| | 10 × 20 pixels | 0.40 | 0.57 | 88.2 | 95.6 | 0.840 |
| | 14 × 28 pixels | 0.42 | 0.57 | 88.8 | 96.9 | 0.857 |
| RVI with HH/HV | 7 × 14 pixels | 0.79 | – | 96.2 | 96.7 | 0.933 |
| | 10 × 20 pixels | 0.79 | – | 96.4 | 97.2 | 0.940 |
| | 14 × 28 pixels | 0.79 | – | 96.7 | 97.5 | 0.946 |
| RVI with VV/VH | 7 × 14 pixels | 0.85 | – | 86.9 | 95.6 | 0.825 |
| | 10 × 20 pixels | 0.85 | – | 87.3 | 96.9 | 0.840 |
| | 14 × 28 pixels | 0.86 | – | 88.7 | 96.9 | 0.857 |

UA: User's accuracy (precision) and PA: Producer's accuracy (recall).

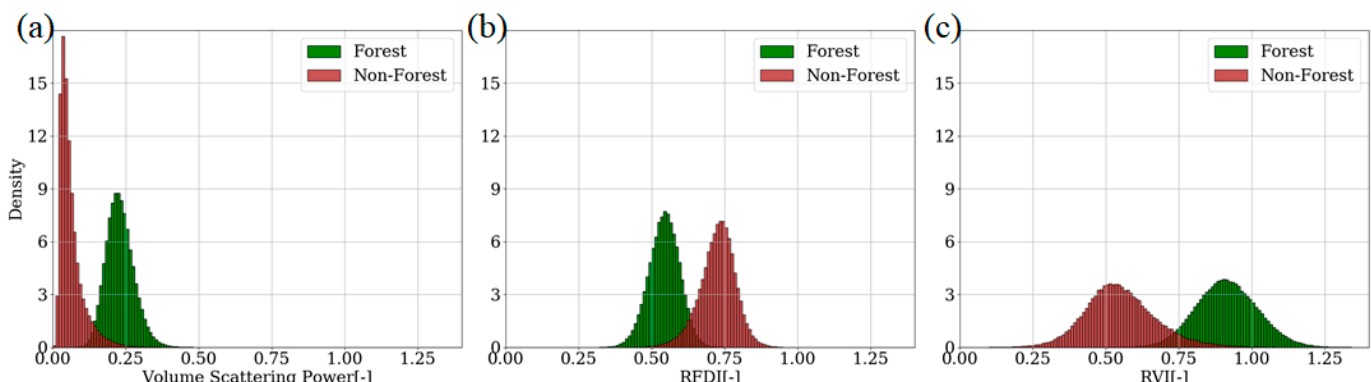

**Figure 6.** Distributions of the (**a**) volume scattering power, (**b**) RFDI, and (**c**) RVI using the forest and non-forest pixels of the reference data. These values were derived using HH/HV data and a window size of $10 \times 20$ pixels.

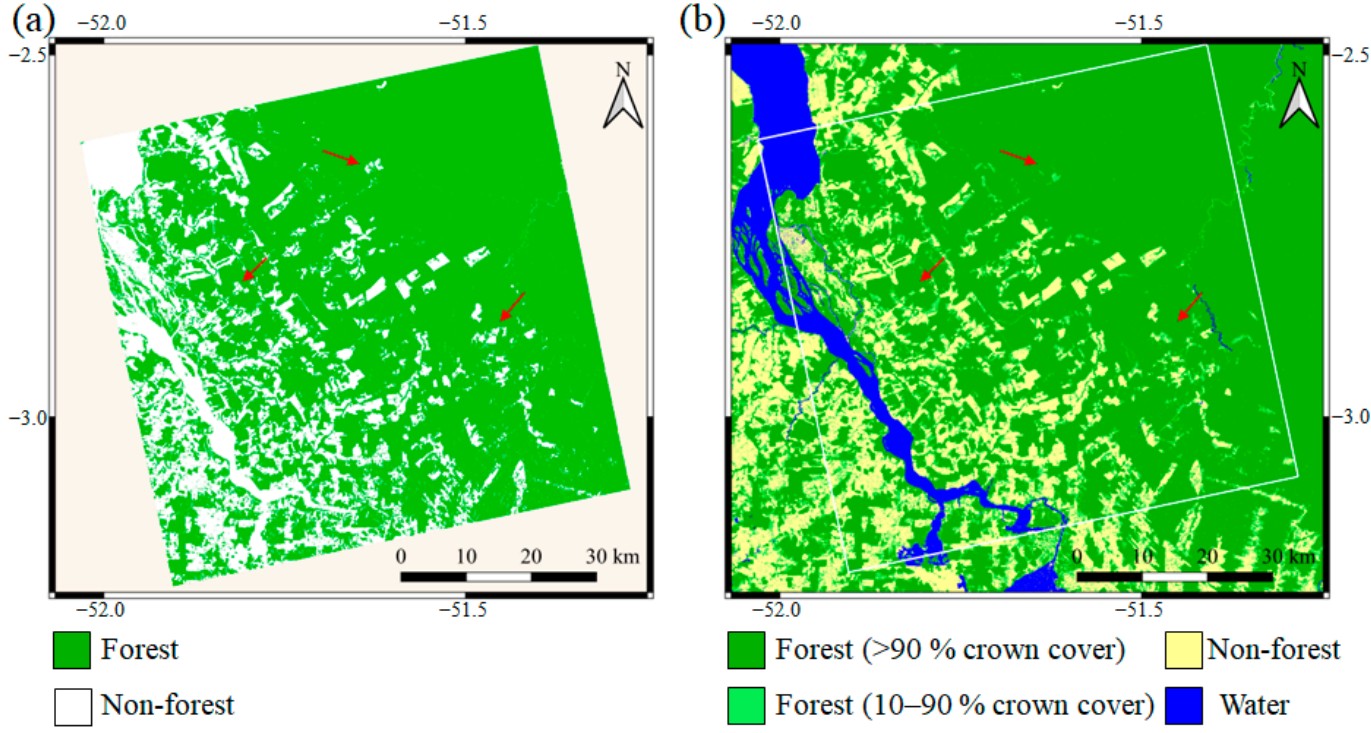

**Figure 7.** Forest map of Altamira using the proposed method with actual dual-polarization data acquired on 4 April 2019. (**a**) Forest map generated by the proposed method. (**b**) JAXA Global PALSAR-2 Forest/Non-Forest map in 2018. The white rectangle shows the observation area of dual-polarization data. The red arrows indicate deforestation in 2019 as visually confirmed by Planet/Dove mosaic images.

The proposed method was applied for deforestation detection at Altamira using actual dual-polarization data acquired on 4 April 2019 and 12 May 2022. The window size was $10 \times 20$ pixels, and the threshold values for $\alpha$ and $\beta$ were set to 0.17 and −0.04, respectively, as given in Table 5. Figures 8 and 9 show examples of deforestation that could and could not be detected by the proposed method, respectively. The detected deforestation area in Figure 8a–c clearly indicated logging and the RGB image based on scattering power (Figure 8d,e) showed a change from green to magenta in the deforested area. This is evidenced by the scattering power components before and after logging in Figure 8e, which shows that the volume scattering power was dominant before logging whereas the ground scattering power increased after logging. In the forest degradation area not detected by the proposed method, Figure 9e,f show that the volume scattering power remained high even after

logging. Figure 9b,c show sparse trees in the area after logging, which indicates that the volume scattering power did not decrease after logging because of the remaining sparse trees. A similar omission error was observed with the 6SD method [8]. Future work will involve improving both methods to detect forest degradation.

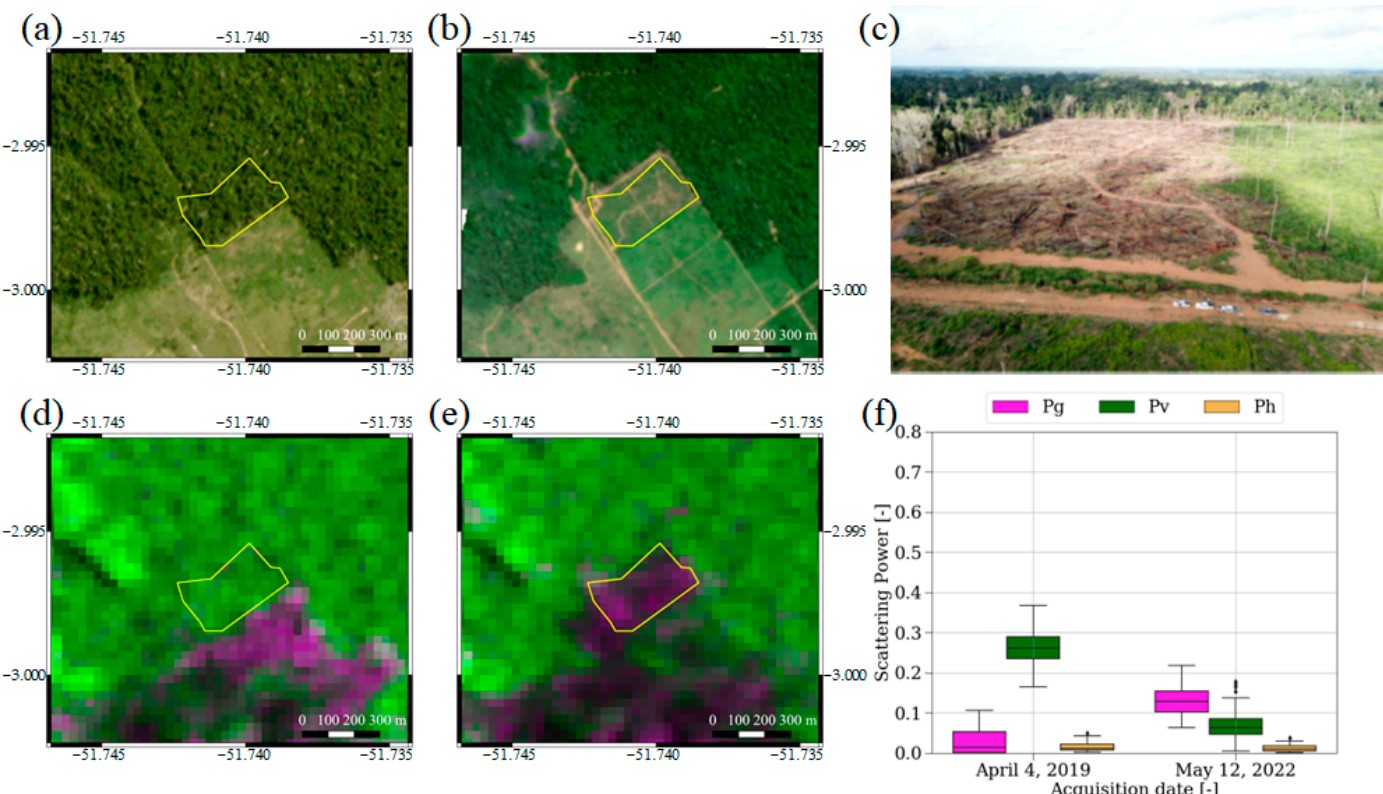

**Figure 8.** Deforestation at Altamira detected by using the proposed method with actual dual-polarization data acquired on 4 April 2019 and 12 May 2022. The yellow outline shows the area detected by the proposed method. Mosaic images acquired by Planet/Dove in (**a**) June–November 2019 and (**b**) April 2022. (**c**) Drone image of the detected area acquired in February 2022. RGB images generated by the proposed method using dual-polarization data acquired on (**d**) 4 April 2019 and (**e**) 12 May 2022. Ground scattering is indicated in red and blue, and volume scattering is in green. (**f**) Box and whisker plot of the scattering power components using pixels in the yellow outline. The whiskers extend to 1.5 times the interquartile range.

## 4. Discussion

The proposed method was evaluated using window sizes of 7 × 14, 10 × 20, and 14 × 28 pixels. The trade-off between spatial resolution of the forest map and accuracy of the forest classification is an important issue for forest monitoring. Table 8 presents the forest classification performance of the proposed method using a window size of 3 × 6 pixels. Figure 10 also shows box and whisker plots of the scattering power components based on the forest pixels of the reference data. The proposed method showed not only comparable UA but also fewer omission errors than the 6SD method at this window size. However, at this window size, the surface and double-bounce scattering powers had approximately the same power as the volume scattering power for the 6SD method with the POLSAR data, whereas the volume scattering power remained higher than the other powers for the proposed method. This suggests that the distribution of scatterers was not uniform in this window size, so the proposed method overestimated the volume scattering power because it assumes that the probability density function only has a uniform distribution. To avoid overestimating the volume scattering power with the proposed method, the window size of the ensemble average should be at least 7 × 14 pixels.

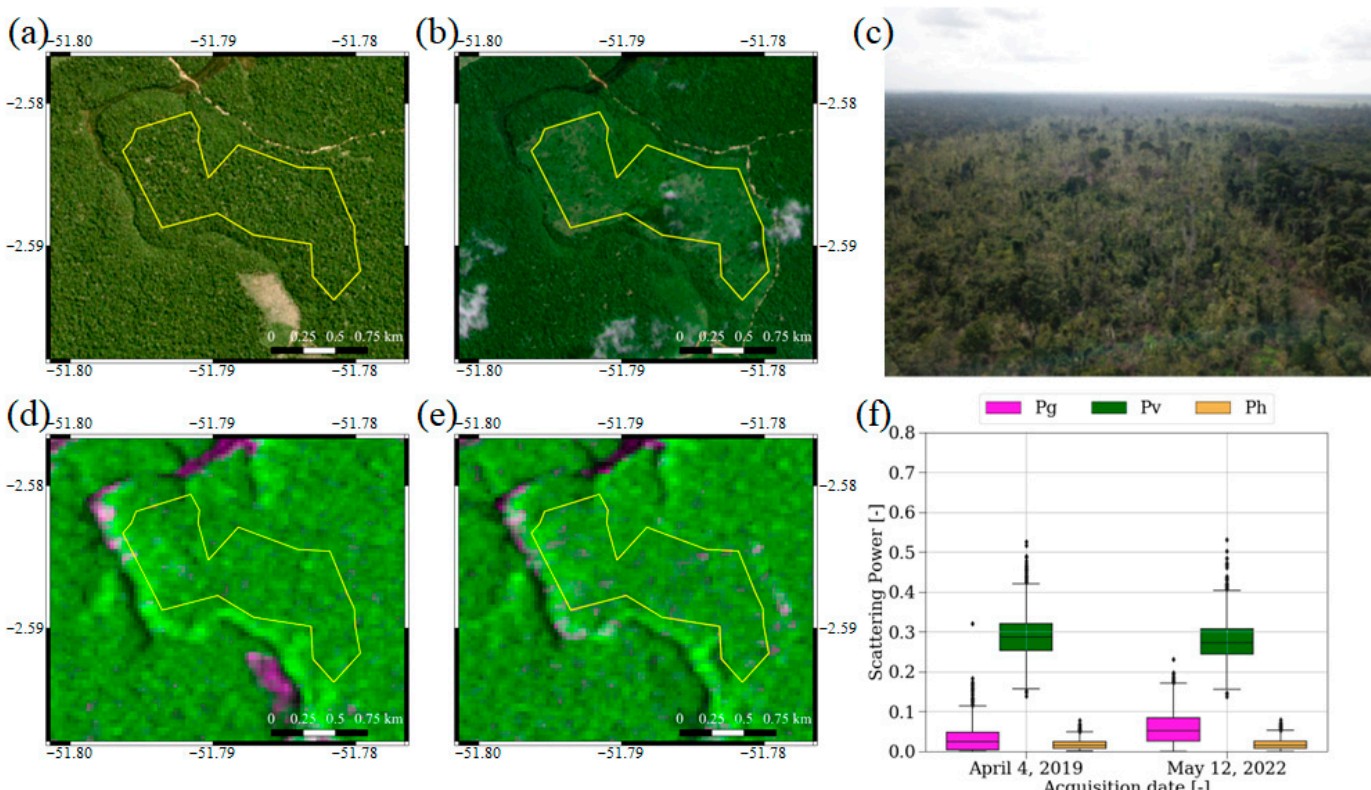

**Figure 9.** Forest degradation at Altamira not detected by using the proposed method with actual dual-polarization data acquired on 4 April 2019 and 12 May 2022. The yellow outline shows the deforestation area based on visual interpretation of Planet/Dove mosaic images. (**a**–**f**) Same as in Figure 8.

**Table 8.** Forest classification performance using a window size of 3 × 6 pixels at Rio Branco.

| Method and Data | Threshold $\alpha$ | UA (%) | PA (%) | Kappa |
|---|---|---|---|---|
| Proposed method with HH/HV | 0.10 | 97.5 | 98.0 | 0.959 |
| Proposed method with VV/VH | 0.11 | 98.0 | 98.2 | 0.964 |
| 6SD method with POLSAR | 0.11 | 97.4 | 59.0 | 0.592 |

UA: User's accuracy (precision) and PA: Producer's accuracy (recall).

The scattering powers generated by the proposed method showed a similar variation to a sigma-naught in mountain regions, where it is stronger at the mountain slope toward SAR line-of-sight and weaker at the mountain slope parallel to SAR line-of-sight. In Figure 4d, the prominent bright pixels at bottom right located at a mountain slope towards the sensor and the dark pixels near those pixels locate at the opposite slope. Because the dark pixels had lower scattering power than that of pixels located at flat area, a part of the mountain forests was not classified as forest in Figure 4e. This suggests that slope correction [23] should be applied to forest classification in high-elevation areas.

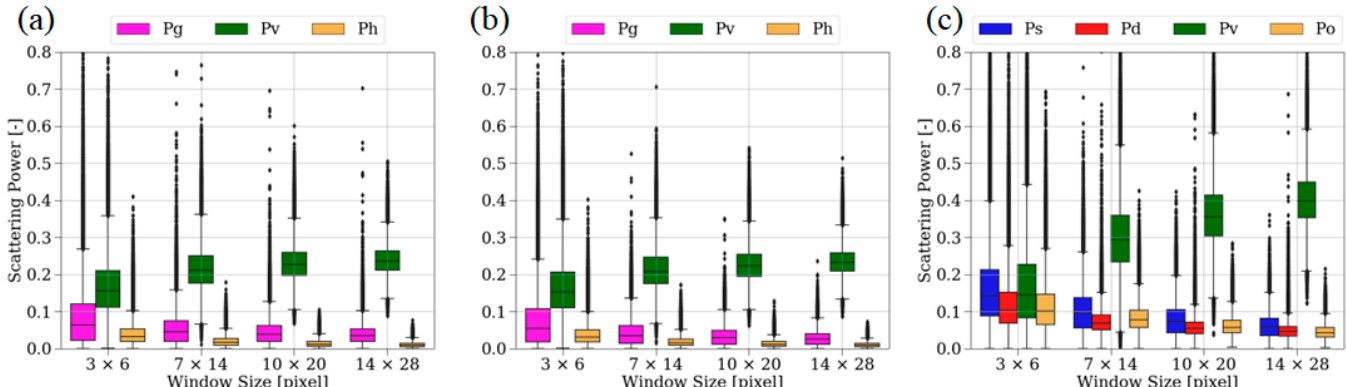

**Figure 10.** Box and whisker plots of the scattering power components using the forest pixels of the reference data: (**a**) proposed method with HH/HV data, (**b**) proposed method with VV/VH data, and (**c**) 6SD method with POLSAR data. The whiskers extend to 1.5 times the interquartile range. In (**c**), $P_o$ is the summation of the helix scattering, oriented dipole scattering, and compound dipole scattering powers.

## 5. Conclusions

We developed a scattering power decomposition method for more accurate monitoring of tropical forest using dual-polarization data. The proposed method decomposes the covariance matrix obtained from dual-polarization data into three scattering powers: ground scattering, volume scattering, and helix scattering. We validated the proposed method by using simulated dual-polarization data. The proposed method showed an excellent forest classification performance with both the UA and PA above 98% at a window size greater than $7 \times 14$ pixels regardless of transmission polarization. It also showed a comparable deforestation detection performance to the 6SD method. The proposed method can contribute to rapid deforestation detection with the same accuracy as the 6SD method because dual-polarization data are observed more frequently. The proposed method showed a better and more robust forest classification performance than the RFDI and RVI regardless of the transmission polarization. When we applied the proposed method to actual dual-polarization data at Altamira, it generated a forest map and demonstrated deforestation detection with high accuracy. The proposed method can be applied to monitoring tropical forests using not only future dual-polarization data but also accumulated data that have not been fully utilized.

Both the proposed method and 6SD method showed omission errors when detecting forest degradation because the volume scattering power did not decrease after logging when sparse trees remained. Future work will involve improving the detection performance of forest degradation by both methods. An application to other SAR data is also planned for future work because the proposed method has no limit to SAR frequency and is applicable to not only L-band data but also to C-band data.

**Author Contributions:** Conceptualization, R.S., R.N., C.T., and Y.Y.; methodology, R.S.; software, R.S.; validation, R.S.; formal analysis, R.S.; investigation, R.S.; resources, R.N. and C.T.; data curation, R.S.; writing—original draft preparation, R.S.; writing—review and editing, R.N., C.T., and Y.Y.; visualization, R.S.; supervision, Y.Y. All authors have read and agreed to the published version of the manuscript.

**Funding:** This article is based on results obtained from the project JPNP20006 commissioned by the New Energy and Industrial Technology Development Organization (NEDO).

**Data Availability Statement:** Not applicable.

**Acknowledgments:** The ALOS-2 original data are copyrighted by JAXA and were provided under the JAXA-AIST agreement. The drone images were acquired by JICA in a field survey on Altamira.

**Conflicts of Interest:** The authors declare no conflict of interest.

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
