# Peer review of "Extension of Scattering Power Decomposition to Dual-Polarization Data for Tropical Forest Monitoring"

_remotesensing, doi:10.3390/rs15030839_

Round 1

Reviewer 1 Report

This paper describes proposing the development of a scattering power decomposition method for monitoring tropical forests using dual-polarized SAR data. The forest/non-forest classification results obtained by the proposed method were absolutely validated by comparison with reference data prepared separately and were also validated relative to the six-component scattering power composition (6SD) method, showing the superiority of the proposed method. However, there are some unclear points in the description as commented below. Hope the authors will consider them in the revised version. 

As the major comments:

#1: In section 2.2, the scattering power decomposition method is described, however, many of the descriptions appear to be methods proposed by published studies. The proposed method should be clearly indicated as the originality of this paper.

In addition, the assumptions and conditions of application assumed in the application of the proposed method should be appropriately indicated.

#2: In section 2.3, the proposed method for logging detection is the classification of forested and non-forested areas, which requires the determination of threshold values α and β. In this study,

a) how was the threshold determined? Were they determined to match the reference data?

b) how do you determine the thresholds when you apply the method to actual data without the reference data?

Explanations are necessary.

#3, In section 3.1 and Tables 2, 4, and 5: The number of evaluation points used in the accuracy evaluation should be indicated. Although the reviewer assumed that the evaluation results were based on the reference data shown in Table 2, it should be clearly indicated. In this case, please clarify the difference between the reference data used to determine the threshold values and the reference data used in the evaluation.

As the minor comments:

#4, L. 84: “an Advanced…” > “the Advanced…”

#5, L. 97: The reviewer thinks the basic observation scenario of ALOS-2 is still ongoing.

#6, L. 335: The term "early detection of deforestation" is misleading, as Line 457 states that "forest degradation is a future task", and early detection may be out of the scope of the subject of this study.

#7, L. 346, Caption of Fig. 2: Please add the window size and how to determine “(c) Forest map” e.g., “(c) Forest map derived by the proposed method”.

#8, L. 352, Figs. 3-5: Please indicate the window size that used individual figures.

#9, L. 397: Does “The reference data were the same ones used in Section 3.1.1.” mean that is summarized in Table 2?

#10, L. 470, Table 8: Why is PA with 6SD inaccurate? Where, it is not only a matter of small window size, as the accuracy of the proposed method with HH/HV and VV/VH is compared using the same PolSAR data. If the volume scattering power is overestimated by the proposed method, it is necessary to argue that it is not affected by different window sizes i.e., larger window size.

Reviewer 2 Report

Monitoring tropical forests is a very important topic, because of the importance of tropical forests for our planet.

This article extended scattering power decomposition from quad-polarized data to dual-polarized data for the purpose of using it in monitoring tropical forests.

Comment 1:

The introduction must be extended o provide sufficient background about the topic.

Comment 2:

It is not mentioned if it is possible to apply the method to other data for example Sentinel1 data (knowing that Sentinel1 data is acquired on C-band). is it the same, and this should be mentioned in the paper.

Comment 3:

The threshold ? used in the method mentioned for the first time in line 247 is not clear, It needs more explanation about the choice of this threshold and if it differs from one site to another.

Comment 4:

The statistical methods are better in the comparative analysis, maybe add histograms to show the difference between the proposed method and other methods.

Round 2

Reviewer 1 Report

The reviewer judges that the peer review comments have been properly reflected.

Please continue to consider quantitative evaluation for accuracy validations.